

# Subjective assessment for super recognition: an evaluation of self-report methods in civilian and police participants

Sarah Bate[1] and Gavin Dudfield[2]

[1] Department of Psychology, Bournemouth University, Poole, UK
[2] Dorset Police, UK

## ABSTRACT

Metacognition about face recognition has been much discussed in the psychological literature. In particular, the use of self-report to identify people with prosopagnosia ("face blindness") has contentiously been debated. However, no study to date has specifically assessed metacognition at the top end of the spectrum. If people with exceptionally proficient face recognition skills ("super-recognizers," SRs) have greater insight into their abilities, self-report instruments may offer an efficient means of reducing candidate lists in SR screening programs. Here, we developed a "super-recognizer questionnaire" (SRQ), calibrated using a top-end civilian sample (Experiment 1). We examined its effectiveness in identifying SRs in pools of police (Experiment 2) and civilian (Experiment 3) participants, using objective face memory and matching tests. Moderate effect sizes in both samples suggest limited insight into face memory and target-present face matching ability, whereas the only predictor of target-absent matching performance across all samples was the number of years that an officer had been in the police force. Because the SRQ and single-item ratings showed little sensitivity in discriminating SRs from typical perceivers in police officers and civilians, we recommend against the use of self-report instruments in SR screening programs.

In the last decade there has been increasing interest in people with extraordinarily proficient face recognition skills—individuals who have become known as "super-recognizers" (SRs, *Russell, Duchaine & Nakayama, 2009*). Identification of this population not only presents a novel theoretical window into the cognitive and neural architecture of the face recognition system (*Bennetts, Mole & Bate, 2017*; *Bobak et al., 2016*, *2017*; *Russell, Duchaine & Nakayama, 2009*), but has also prompted interest into the deployment of SRs in policing and security settings (*Bate et al., 2018*; *Phillips et al., 2018*). Alongside intense media coverage, this surge of interest in super recognition has resulted in large numbers of people self-referring to laboratories in the belief that they possess extraordinary face recognition skills (*Bate et al., 2018*). While there are clear advantages of increased sample sizes for both theoretical and applied purposes, important questions remain about the most efficient and accurate means of screening these individuals.

Corresponding author
Sarah Bate,
sbate@bournemouth.ac.uk

Super-recognizers are typically identified using objective tests of face memory, such as the extended form of the Cambridge Face Memory Test (CFMT+: *Russell, Duchaine & Nakayama, 2009*), or a variety of face matching tests (*Bobak, Dowsett & Bate, 2016*; *Bobak, Hancock & Bate, 2016*; *Robertson et al., 2016*). In policing settings, SRs have also been selected via scrutiny of on-the-job performance (*Phillips et al., 2018*). The latter approach is problematic: it not only precludes the identification of potentially valuable new recruits, but is also confounded by occupational role (and therefore opportunity to demonstrate one's skills) and familiarity with repeat offenders (where the relatively easier task of familiar face recognition is given equal weight to the more challenging task of unfamiliar face recognition, for example, see *Young & Burton, 2017*). On the other hand, screening large numbers of people with objective tests can be time-consuming and may heavily drain resources—particularly in light of recent evidence indicating that repeated assessment is necessary to assess consistency of performance in SR candidates (*Bate et al., 2018*; see also *Bindemann, Avetisyan & Rakow, 2012*).

An alternative is to initially ask people whether they think they have superior face recognition skills, and to subsequently carry out objective screening only with those who return high self-ratings. However, there is mixed evidence in the psychological literature about meta-cognition and face recognition performance, resulting in an enduring and contentious debate about the utility of self-report. Earlier studies used single-item ratings of general face recognition abilities, finding only small-to-moderate correlations with performance on objective face recognition tests (*Bindemann, Attard & Johnston, 2014*; *Rotshtein et al., 2007*). More recently, multi-item questionnaires have been developed that aim to quantify people's experiences of specific behaviors that are associated with prosopagnosia (*Palermo et al., 2017*; *Shah et al., 2015a*; *Stollhoff et al., 2011*; see also *Murray et al., 2018*). While effect sizes have varied substantially in these studies, Shah et al.'s questionnaire elicited relatively stronger effects that persisted through multiple validation studies (*Gray, Bird & Cook, 2017*)—presenting a potential avenue for self-report in the early stages of prosopagnosia screening.

Only one study to date has examined whether this approach may be similarly useful for the identification of SRs. *Bobak, Mileva & Hancock (in press)* adapted some of the items in *Shah et al.'s (2015b)* questionnaire to make the instrument suitably calibrated for use across the full face recognition spectrum. They found only moderate associations ($r = 0.32$) with face recognition performance in naïve typical participants (i.e., those who had no objective knowledge about their face recognition skills). While a group of SRs more accurately rated their face recognition abilities, these individuals had previously been informed of their objectively-confirmed SR status. The authors included these participants to demonstrate that prior-knowledge of top-end performance can inadvertently increase effect sizes. However, it remains unknown whether self-report can accurately identify *naïve* SRs. A recent report partly addresses this question: objective screening of 200 people who believed they are SRs revealed that 59.5% of the sample met the most liberal inclusion criteria for super recognition, although this figure dropped to 2.5% when consistency of performance was also taken into account (*Bate et al., 2018*). It is possible that a behavioral trait questionnaire that is specifically calibrated to tap

top-end performance will result in a reduced short-list for SR screening compared to a simple self-referral system, although it is unclear whether such an instrument will also fail to detect some SRs.

Another issue that has not yet been examined is the use of self-report to identify SRs in the police force. It is possible that some police officers may have more accurate insights into their face recognition skills compared to civilians, because they receive additional opportunities to directly scrutinize their face recognition ability (i.e., when matching faces captured in CCTV footage). However, it is also possible that these opportunities elicit a different level of calibration for self-report in police participants: while civilians may rate their skills according to everyday familiar face recognition performance (i.e., recognizing the faces of family members and friends, where errors are seldom made), police officers are often required to consider the faces of unfamiliar individuals (i.e., when deciding if two facial images match in identity, or when searching a crowd or CCTV footage for a suspect or missing person). In both scenarios, two faces may match in identity (known as "target-present" instances), or they may be two different people ("target-absent" instances). Notably, existing work has not only dissociated face memory from face matching performance in some SRs, but also target-present from target-absent accuracy (*Bate et al., 2018*; *Bobak, Hancock & Bate, 2016*). Whether these more intricate measures of top-end face recognition ability can also be detected via self-report is another important outstanding question.

The current study investigated these issues. We developed a new 20-item "super recognizer questionnaire" (SRQ) that enquired about everyday face recognition experiences that were frequently described in previous informal discussions with objectively-identified SRs (e.g., those participating in our laboratory's previously published work: *Bate et al., 2018*; *Bobak, Pampoulov & Bate, 2016*). Experiment 1 validated the SRQ using a large sample of civilian participants who believed they had superior face recognition skills, but had never taken part in objective assessments. Importantly, performance on two objective face recognition tests (assessing face memory and face matching, with the latter containing target-present and target-absent trials) was collected *after* the participants had completed the SRQ—ensuring that they could only draw upon their everyday experiences when completing the questionnaire. In a second experiment, we addressed the same issues in a sample of police officers, who had not been pre-selected according to their self-perceived face recognition skills. To investigate whether occupational pressures or on-the-job experiences influenced self-report of face recognition ability in these individuals, a final experiment compared their performance to a sample of typical civilian participants.

## EXPERIMENT 1

An initial experiment validated the SRQ in a large sample of citizens who had self-referred to our laboratory in the belief that they are SRs, but had not previously taken part in any objective face recognition tests. The benefits of using this sample were threefold: Investigation of participants with above-average face recognition skills permitted the validity of the behavioral traits used in the SRQ to more sensitively be examined; all participants were naïve about their "true" SR status, preventing any objective

information from influencing their self-ratings; and the full anonymity and independence of the study from organizational pressures or outcomes encouraged honest responding.

## METHOD

### Participants

A total of 264 (181 female) Caucasian civilians took part in this study. They were aged between 18 and 50 years ($M$ = 37.2, SD = 7.7). Following media coverage of our previous work, all participants had registered their details on our laboratory's website (www.prosopagnosiaresearch.org), expressing their interest in participation in an online SR screening program. No participant declared prior participation in screening tests that had been run by other laboratories, and all were advised that no occupational opportunities would arise from the outcomes of the study. Ethical approval (application ID 11487) was granted by Bournemouth University's Ethics Committee, and written informed consent was collected from all participants.

### Materials

A 20-item SRQ was developed (see Table 1). Each question asked participants to rate their face recognition skills in a given context, using a Likert scale of 1–5 (where for half the items 1 represented "strongly disagree" and 5 represented "strongly agree"; the remaining items were reverse-coded). Questions were developed following informal discussions with existing (objectively-confirmed) SR research participants. To ensure content validity, the questions were designed to probe different aspects of face-processing, contextualized within everyday scenarios. For instance, some enquired about face memory, others about face matching, and the remainder about "spotting" faces in a crowd. After responses were modified to account for reverse coding, ratings for each item were summed to give a total score out of 100, where higher scores corresponded to better face recognition skills.

Participants also completed two objective tests of face-processing, suitably calibrated to detect top-end performance. Face memory was measured using the extended form of the (CFMT+, *Russell, Duchaine & Nakayama, 2009*). This popular test has been used in all SR investigations reported to date, and is described elsewhere (see *Russell, Duchaine & Nakayama, 2009* for full details). In brief, participants are required to learn six faces in an initial encoding stage: they then select each target from three test items, each containing the relevant target and two distractors. After reviewing the six targets for 20 s, participants are required to select a target from 30 additional triads of faces, now presented under novel lighting or viewpoint conditions. Participants then review the targets for a further 20 s, before completing 54 more difficult trials, some with different facial expressions and added noise.

Face matching skills were measured using the pairs matching test (PMT; *Bate et al., 2018*). This task contains 48 trials: 24 match in identity and the remainder display two different individuals. All images were downloaded from Google image searches, and were cropped to display the entire face from the neck upward. Mismatched faces were paired according to their perceived similarity to each other, and all images were adjusted to 10 cm in width and 14 cm in height. Stimuli were displayed in a random order until

| Table 1 PCA loadings for each item on the SRQ. | | | |
|---|---|---|---|
| Item | Factor 1 | Factor 2 | Factor 3 |
| I cannot recognize familiar people when their hair is covered by a hat or hood. | | | |
| I can tell when two people are related just by looking at their faces. | | 0.24 | 0.64 |
| I cannot recognize the faces of people who I have only seen once before. | 0.40 | | |
| When meeting a new person at a pre-arranged spot I often struggle to find them despite having seen their photograph. | | 0.30 | |
| I find it difficult to intentionally locate a familiar face in a crowd. | | | |
| I am better at face recognition than most other people. | 0.26 | 0.79 | |
| I can recognize the faces of actors when they have substantially aged. | | 0.68 | 0.38 |
| I struggle to know when two photographs taken a long time apart are of the same person. | | | 0.64 |
| I can spot familiar people in unexpected contexts. | | 0.77 | |
| I cannot recognize the faces of people who I have not seen since childhood. | | | 0.37 |
| I never notice famous faces in unexpected locations or images. | | | |
| I am worse at face recognition than my closest family or friends. | | 0.22 | |
| I can recognize unknown actors playing minor roles across different television program. | 0.66 | | |
| I can recognize familiar people from their childhood photographs. | 0.50 | | 0.62 |
| I have previously recognized someone who didn't recognize me. | 0.68 | 0.27 | |
| I know when two poor quality photographs are of the same person. | 0.59 | | 0.49 |
| Crowds of faces look the same to me. | 0.40 | 0.38 | |
| I am known amongst my friends and/or family for my good face recognition skills. | 0.69 | | |
| I think all babies look the same. | | | 0.21 |
| I sometimes spot people that I don't know well in a crowd. | 0.51 | 0.55 | |

responses were made, and no time limit was imposed. The proportion of hits and correct rejections were independently summed for this task.

## Procedure

Due to the large sample size and varied geographical locations of the participants, all data were collected online. Participants were initially asked to complete a questionnaire that enquired about their demographic background and previous participation in face recognition studies. They also answered two stand-alone questions about their face recognition skills: First, they were asked to rate their ability on a Likert scale that ranged

from 1 to 5 (where 1 represented "very poor" and 5 "very good"); second, a close family member or friend made the same rating about the participant's abilities (these ratings are subsequently referred to as "single-item self-ratings" and "single-item other-ratings," respectively). These questions were included to examine whether a multi-item trait questionnaire improved upon more general single-item ratings, provided by either by oneself or a close other.

Participants then completed the SRQ, followed by the CFMT+ and the PMT. The demographic questionnaire and SRQ were always completed first and in the same order; presentation of the CFMT+ and PMT was counterbalanced between participants. Technical errors were monitored by the website (e.g., interruptions in Internet connection during test completion). Participants also completed a follow-up questionnaire that enquired about technical issues and whether they had received assistance with any part of the process.

## RESULTS

### Validity

An exploratory principal components analysis (PCA) with Varimax rotation was initially carried out on data collected from the 20 items of the SRQ. Three factors were identified that had Eigenvalues greater than 1.3, and collectively explained 42.39% of the variance (see Table 1). The first factor explained 27.35% of the variance and loaded more heavily on items that tap face memory. The second explained 8.23% of the variance and contained items that assess the "spotting" of faces in a crowd. The final factor explained 6.81% of the variance and mostly contained items that correspond to face matching, particularly from photographs. The SRQ had very good internal reliability: Cronbach's $\alpha$ was 0.85, and the split-half Spearman–Brown coefficient was 0.79. The size of these values suggest that all items are worthy of retention. Item-analyses did not reveal any large increases in reliability following the removal of individual questions, nor were there any gains in creating sub-scales according to the results of the PCA. Thus, we retained all items in the questionnaire for the remaining analyses.

### Sensitivity

Performance on the objective tests (CFMT+ and PMT) was used to infer the members of the sample who met the criteria for super recognition: scores that exceeded 1.96 SDs above control cut-offs on both tests, using existing norms (*Bate et al., 2018*; see Table 2). According to these criteria, 26.9% of the sample ($N = 71$, 54 female) were deemed to be "SRs." Even though the sample contained mostly above-average performers, a between-groups MANOVA on the three subjective measures (SRQ, single-item self-rating, single-item other-rating) revealed a statistically significant difference in the overall model between confirmed SRs and the remainder of the sample (hereon referred to as "typical perceivers"), $F(3,260) = 2.754$, $p = 0.043$, partial $\eta^2 = 0.031$. SRs rated their face recognition skills more highly than typical participants on the SRQ, $F(1,262) = 7.834$, $p = 0.006$, partial $\eta^2 = 0.029$, but not the single-item self- or other-ratings, $F(1,262) = 2.554$, $p = 0.111$, and $F(1,262) = 1.540$, $p = 0.216$, respectively.

**Table 2 Overall mean (SD) of scores on all tests in each experiment.**

| | SRQ | CFMT+ | PMT: All | PMT: TP | PMT: TA |
|---|---|---|---|---|---|
| Existing norms (*Bate et al., 2018*) | N/A | 68.16 (9.94) | 68.80 (7.36) | 67.92 (17.12) | 69.69 (16.42) |
| Top-end civilians (Exp 1) | 89.64 (8.11) | 84.22 (9.36) | 80.23 (8.32) | 78.60 (14.06) | 81.87 (13.42) |
| Non-selected police officers (Exp 2) | 78.91 (9.94) | 73.84 (11.55) | 74.69 (9.15) | 76.68 (14.23) | 72.71 (14.77) |
| Non-selected civilians (Exp 3) | 65.93 (9.79) | 64.30 (13.42) | 65.69 (9.76) | 66.17 (14.84) | 65.21 (16.45) |

**Note:**
Note that higher scores in Experiment 1 reflects the greater proportion of SRs in this sample, and more SRs were also identified in Experiment 2 than Experiment 3.

**Table 3 Mean (SD) and range of subjective face recognition scores for the 71 SR and 193 typical (civilian) participants reported in Experiment 1.**

| | SRQ | Single-item self-rating | Single-item other-rating |
|---|---|---|---|
| SRs | 91.92 (6.89) | 4.54 (0.50) | 4.63 (0.54) |
| | 69–100 | 4–5 | 3–5 |
| Typical participants | 88.80 (8.38) | 4.42 (0.50) | 4.53 (0.60) |
| | 57–100 | 4–5 | 3–5 |

It is also notable that the SRs' SRQ scores ranged from 69 to 100, whereas the typical participants' scores ranged from 57 to 100. While there is greater variance in SRQ scores for both SR and typical participants compared to single-item scores (see Table 3), the overlap in SRQ scores between the two groups is considerable (see Fig. 1). A total of 15 SRs returned SRQ scores that were at least 1 SD below the SR mean (three of these individuals returned scores that were at least 2 SDs below the mean). Further, 88 typical participants returned SRQ scores that were above the SR mean. While this is not surprising given the sample all self-referred for super recognition, the CFMT+ scores achieved by 60 of these individuals were at least 2 SDs below the SR mean.

## Relationship with objective measures

Multiple regression analyses were performed to investigate whether subjective ratings (single-item self-rating, single-item other-rating and SRQ) significantly predicted participants' CFMT+ and PMT performance (see Fig. 1). The results of the first regression indicated that the model explained 7.8% of the variance, and was a significant predictor of CFMT+ performance, $F(3,260) = 7.350$, $p = 0.001$. While the SRQ significantly predicted CFMT+ scores ($\beta = 0.214$, $p = 0.001$), neither single-item self- ($\beta = 0.081$, $p = 0.269$) nor other- ($\beta = 0.017$, $p = 0.817$) ratings contributed to the model.

A second multiple regression was carried out to see if the same independent variables predicted overall scores on the PMT. The model explained 6.6% of the variance, and significantly predicted performance, $F(3,260) = 6.124$, $p = 0.001$. Both the SRQ ($\beta = 0.187$, $p = 0.006$) and single-item other-rating ($\beta = 0.164$, $p = 0.028$) were significant predictors, but not the single-item self-rating ($\beta = -0.128$, $p = 0.074$). To examine whether target-present and target-absent face matching performance were differentially related to the self-report measures, we carried out two further regressions. The target-present

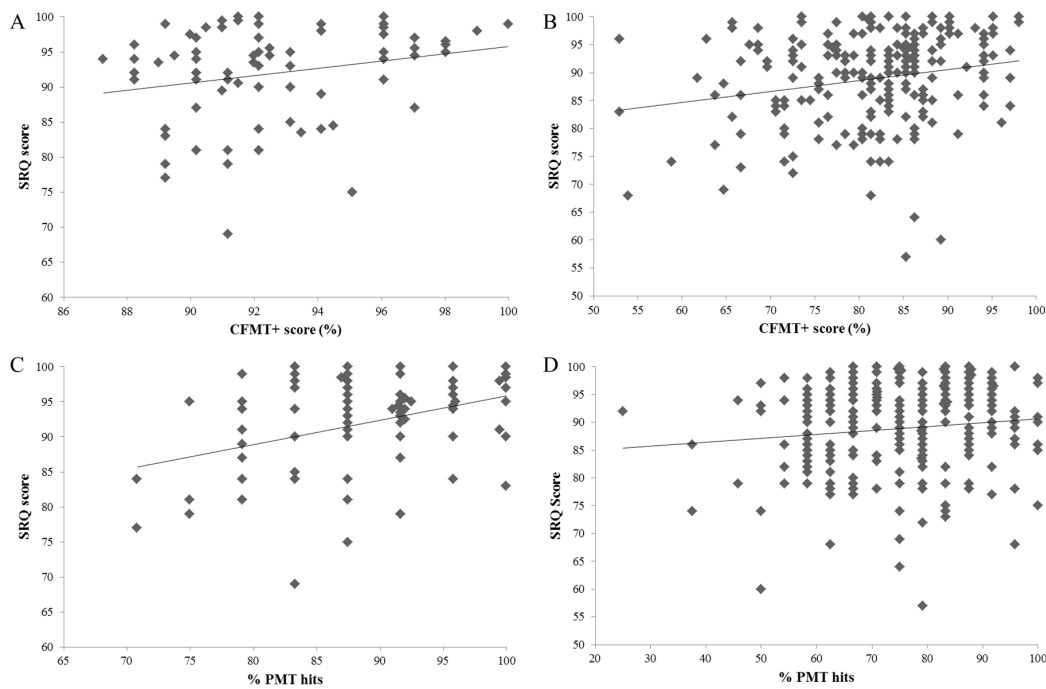

**Figure 1 The relationship between SRQ scores and objective face recognition performance in super-recognizer and typical civilian participants.** (A) Relationship between SRQ and CFMT+ scores in super-recognizer participants. (B) Relationship between SRQ and CFMT+ scores in typical civilian participants. (C) The association between SRQ and target-present face matching performance (hits) for super-recognizers. (D) The association between SRQ and target-present face matching in typical participants.

model explained 4.8% of the variance, and significantly predicted performance, $F(3,260) = 4.370$, $p = 0.005$. However, only the SRQ was a significant predictor ($\beta = 0.174$, $p = 0.012$), and not the single-item self- ($\beta = -0.055$, $p = 0.442$) nor other- ($\beta = 0.106$, $p = 0.156$) ratings. The target-absent model explained only 1.2% of the variance, and did not significantly predict performance, $F(3,260) = 1.028$, $p = 0.381$.

## Correlations by group

Finally, we examined whether performance by either SRs or typical participants might be driving any overall significant associations between subjective and objective performance (see *Bobak, Mileva & Hancock, in press*). While all three subjective ratings were significantly associated with CFMT+ performance in overall correlations, correlation co-efficients were remarkably similar for SRQ and single-item other-ratings in SR and typical participants (see Table 4). In contrast, larger correlations were observed in SRs compared to typical participants for the target-present trials of the PMT; and Fisher *r*-to-*z* transformations found the difference in the size of the correlations to be significant for the SRQ ($z = 1.97$, $p = 0.049$) and single-item self-rating ($z = 2.48$, $p = 0.013$). However, little evidence for accurate insight into target-absent PMT performance was observed in either group. Notably, mild but negative effects were observed for the SRQ and single-item self-ratings in SR but not typical participants; the reduced ability of self-report measures to

**Table 4 Correlations between subjective and objective face recognition scores for the 71 SR and 193 typical (civilian) participants reported in Experiment 1.**

|  | CFMT+ | PMT: TP | PMT: TA |
|---|---|---|---|
| SRQ |  |  |  |
| SRs | 0.22 | 0.37** | −0.25* |
| Typical | 0.21* | 0.11 | 0.06 |
| All | 0.26** | 0.20** | 0.05 |
| Self-rating |  |  |  |
| SRs | −0.01 | 0.31* | −0.19 |
| Typical | 0.15* | −0.03 | −0.04 |
| All | 0.16* | 0.06 | −0.04 |
| Other-rating |  |  |  |
| SRs | 0.20 | 0.32* | −0.05 |
| Typical | 0.19* | 0.11 | 0.06 |
| All | 0.20** | 0.16* | 0.06 |

Notes:
* $p < 0.05$,
** $p < 0.001$.

discriminate between SR compared to typical perceivers was confirmed for the SRQ via a significant Fisher $r$-to-$z$ transformation ($z = 2.23$, $p = 0.026$).

## Summary

While the SRQ fared better than either of the single-item ratings (particularly in SR compared to typical participants), mild effect sizes in all participants suggest that the instrument may have limited use in practical settings. Although a mild relationship was also observed for target-present face matching performance, the SRQ did not accurately predict target-absent face matching performance.

## EXPERIMENT 2

Having validated the SRQ in a civilian sample, our second experiment explored whether the instrument can be used to identify potential SRs in a formal occupational screening program within the police force. To examine whether professional experience can aid either subjective or objective performance, we also took account of the number of years that each officer had worked for the police.

## METHOD

### Participants

A total of 151 Caucasian police officers (100 male) participated in this study. They were aged between 20 and 50 years ($M = 37.5$, $SD = 7.1$), and had worked as police officers for 0–31 years ($M = 11.0$, $SD = 6.7$). Officers responded to an open call for the screening program, where advertisements urged participation regardless of self-perceptions of face recognition ability. They were assured that no feedback on any individual's performance would be released to the organization, although the identity of any confirmed

**Table 5 Mean (SD) and range of subjective face recognition scores for the 10 SR and 141 typical police officers reported in Experiment 2.**

|  | SRQ | Single-item self-rating | Single-item other-rating |
|---|---|---|---|
| SRs | 79.70 (10.63) | 3.80 (0.42) | 3.89 (0.60) |
|  | 60–94 | 3–4 | 3–5 |
| Typical participants | 78.86 (9.9) | 3.67 (0.72) | 3.62 (0.81) |
|  | 48–100 | 2–5 | 1–5 |

Note:
   Note that single-item other-ratings were not provided by one SR and 16 typical officers.

SRs could be presented with each person's permission. Ethical approval was granted by the institutional Ethics Committee.

## Materials and procedure

The same materials and procedure were used as in Experiment 1. Single-item ratings from "others" were provided by colleagues (15 officers did not provide a response to this question, but all completed the SRQ and provided the single-item self-rating). Both the CFMT+ and PMT were completed by 94 officers, 42 only completed the CFMT+, and 15 only the PMT. All data were retained to increase the power of the analyses.

## RESULTS

### Validity

The SRQ continued to show excellent internal reliability: in this sample Cronbach's α was 0.90, and the split-half Spearman–Brown coefficient was 0.87.

### Sensitivity

Using the same parameters as Experiment 1, 10 officers were subsequently deemed to be SRs based on their CFMT+ and PMT performance (see Table 2). Four further individuals achieved scores that were above the cut-off in one test (two on the CFMT+ and two on the PMT), but did not complete the second test. As the scores achieved by all four officers were very close to the cut-off (and these individuals may only be borderline cases for super recognition), we did not include them in the SR sample.

A between-groups MANOVA on the three subjective measures did not elicit a statistically significant difference in the overall model between confirmed SRs and typical perceivers, $F(3,130) = 0.362$, $p = 0.781$, although the mean ratings on each measure were numerically higher for the SR group (see Table 5). While this null result may be attributed to a lack of power in the MANOVA, even independent-samples $t$-tests on the three subjective measures were far from significance: $t(10,141) = 0.909$, $p = 0.380$ (single-item self-rating), $t(9,125) = 1.244$, $p = 0.241$ (single-item other-rating), $t(10,141) = 0.258$, $p = 0.797$ (SRQ).

The SRQ demonstrated little sensitivity in discriminating between SRs and typical perceivers. SR scores ranged from 60 to 94 ($M = 79.70$, SD = 10.63), whereas the scores of typical officers ranged from 48 to 100 ($M = 78.86$, SD = 9.90). One officer scored 100% on

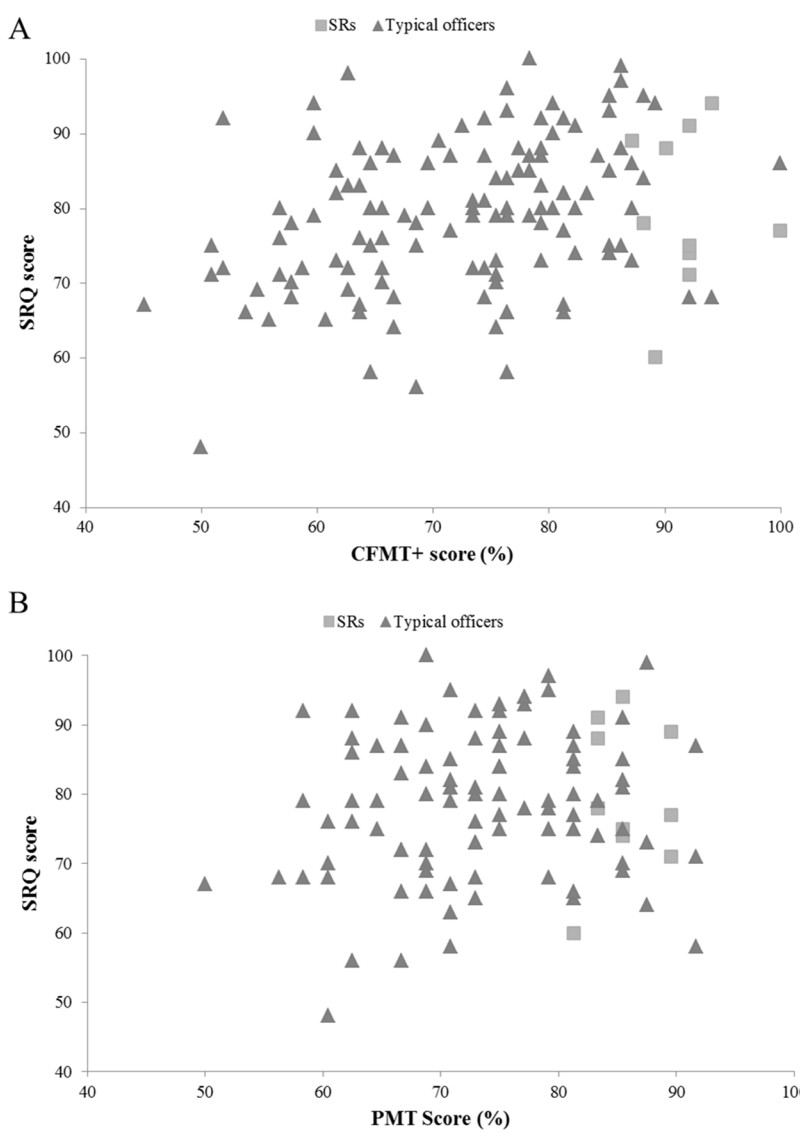

**Figure 2 The relationship between SRQ and objective face recognition performance in super-recognizer and typical police officers.** (A) The relationship between SRQ and CFMT+ scores for super-recognizer and typical police officers. (B) The relationship between SRQ and target-present face matching scores for super-recognizer and typical police officers.

the CFMT+, yet only returned a SRQ score of 77. Very similar patterns were observed for the two single-item ratings (see Fig. 2; Table 5).

## Relationship to objective measures

Multiple regression analyses were again used to assess whether subjective ratings (single-item self-rating, single-item other-rating and SRQ scores), and the number of years that each officer had been in the police force, predicted objective performance on the two face recognition tests. The first regression examined the effectiveness of these predictors against percentage accuracy on the CFMT+: the model explained 19.3% of the

**Table 6 Correlations between subjective and objective face recognition scores for the 10 SRs and 141 typical police officers reported in Experiment 2.**

|  | CFMT+ | PMT: TP | PMT: TA |
|---|---|---|---|
| SRQ |  |  |  |
| SRs | 0.03 (10) | −0.21 (10) | 0.38 (10) |
| Typical | 0.37** (126) | 0.12 (99) | 0.03 (99) |
| All | 0.32** (136) | 0.10 (109) | 0.05 (109) |
| Self-rating |  |  |  |
| SRs | −0.42 (10) | −0.27 (10) | 0.39 (10) |
| Typical | 0.11 (126) | 0.19 (99) | −0.14 (99) |
| All | 0.11 (136) | 0.18 (109) | −0.10 (109) |
| Other-rating |  |  |  |
| SRs | −0.13 (9) | −0.40 (9) | 0.34 (9) |
| Typical | 0.33** (112) | 0.29* (85) | −0.01 (85) |
| All | 0.32** (121) | 0.27* (94) | 0.04 (94) |
| Time in police |  |  |  |
| SRs | −0.26 (10)* | −0.20 (10) | −0.07 (10) |
| Typical | −0.14 (126) | −0.08 (99) | 0.20* (99) |
| All | −0.09 (136) | −0.07 (109) | 0.21* (109) |

Notes:

Note that single-item other-ratings were not provided by one SR and 16 typical officers. Both the CFMT+ and PMT were completed by 94 officers, 42 only completed the CFMT+, and 15 only the PMT. Sample size for each correlation is presented in parentheses.

**$p < 0.001$, *$p < 0.05$ (note that these correlations are non-significant when a correction for multiple comparisons is applied).

variance, and was a significant predictor of CFMT+ performance, $F_{(4,116)} = 6.927$, $p = 0.001$. Both the SRQ ($\beta = 0.282$, $p = 0.009$) and single-item other-ratings ($\beta = 0.342$, $p = 0.008$) significantly predicted performance. Single-item self-ratings had a significant but negative effect ($\beta = −0.256$, $p = 0.043$), and there was no influence of the length of time that a participant had been in the police ($\beta = −0.131$, $p = 0.123$).

A second multiple regression used the same predictors to produce a model that explained 9.2% of the variance in target-present performance (percentage accuracy) on the PMT, but did not reach significance, $F_{(4,89)} = 2.263$, $p = 0.069$. Finally, a regression was carried out on PMT target-absent scores (percentage accuracy), using the same predictors. This model explained 11.9% of the variance, and significantly predicted performance, $F_{(489)} = 2.999$, $p = 0.023$. Years in the police force significantly predicted performance ($\beta = 0.230$, $p = 0.025$). Single-item self-ratings had a negative but significant effect ($\beta = −0.376$, $p = 0.011$). Neither the SRQ ($\beta = 0.227$, $p = 0.092$) nor single-item other-ratings ($\beta = 0.142$, $p = 0.340$) contributed to the model.

## Group analyses

Individual correlations for SRs and typical officers were also performed. Correlations for typical participants supported the findings of the multiple regression analyses (see Table 6). Because of the small sample size in the SR group ($N = 10$), analyses for that

group alone were not deemed to be particularly meaningful. Interestingly, their inclusion in overall analyses did not inflate effect sizes.

## Summary

In non-SR officers, the SRQ was only a significant predictor of CFMT+ and not matching performance. Single-item self-ratings had a negative relationship with face memory and matching scores, suggesting they should particularly be avoided. There may be more utility in requesting SR nominations from colleagues, as single-item other ratings were a good predictor of CFMT+ performance. Importantly, the length of time that an officer has been in the police force was only found to assist target-absent face matching performance.

# EXPERIMENT 3

While Experiment 2 found little support for use of the SRQ in policing settings, it is unclear why effects were smaller than those reported in Experiment 1. It is possible that the different patterns of findings result from the differences in self-perceived face recognition ability between the two samples (i.e., the civilian participants in Experiment 1 all believed that they were SRs, whereas the police officers in Experiment 2 were invited to participate in the study regardless of their self-perceived face recognition ability). This possibility may also reflect more genuine differences in objective face recognition ability between the two groups. Alternatively, it may be that police officers are subject to certain occupational pressures or experiences that make them less accurately self-report their face recognition skills. To address this issue, our final study administered the SRQ, CFMT+ and PMT to a randomly-selected civilian sample.

# METHOD

## Participants

A total of 100 Caucasian civilian participants (38 female) participated in this study, aged between 18 and 46 years ($M$ = 26.3 years, SD = 6.7). They were recruited via Prolific—an online research participant recruitment database (www.prolific.ac). Ethical approval was granted by the institutional Ethics Committee.

## Materials and procedure

Participants initially completed the SRQ, followed by the CFMT+ and PMT, as described for the previous two experiments.

# RESULTS

Using the same parameters as Experiment 1, two individuals were deemed to be SRs based on their CFMT+ and PMT performance (see Table 2). Because this sample size is too small for further analyses, we excluded these individuals from the sample and performed a series of correlations to assess the relationship between the SRQ and the three objective measures (regression analyses were not performed as we only had one measure of self-report in this population). No significant correlation was observed between the SRQ and the CFMT+ ($r$ = 0.16, $p$ = 0.117), nor between the SRQ and

target-present ($r = 0.08$, $p = 0.430$) or target-absent ($r = -0.04$, $p = 0.685$) performance on the PMT. These findings indicate that the SRQ is better-calibrated to distinguish between top-end performers in all participants, regardless of occupational status.

## DISCUSSION

This investigation examined the utility of subjective measures in predicting objective face recognition performance in self-referred civilian SRs (Experiment 1), typical police officers (Experiment 2), and typical civilian participants (Experiment 3). A new self-report questionnaire (the SRQ) that aimed to quantify behavioral traits of super recognition was found to have high internal reliability. In top-end civilian participants, the SRQ was a better (but still only moderate) predictor of face memory and target-present face matching performance than a single-item self-rating, whereas very little statistical support was found for the use of self-report in typical police officers or civilians.

Akin to existing work that has examined self-report at the other end of the face recognition spectrum (i.e., in those with developmental prosopagnosia: *Shah et al., 2015b*), our findings indicate that a behavioral trait questionnaire is a better predictor of face recognition performance in top-end performers than a more generalized single-item self-rating. In civilian top-end participants, this finding held for both face memory and target-present face matching, although more intricate patterns emerged when SR and typical participants' performance were independently analyzed. While effect sizes for SR and typical participants were remarkably similar for face memory correlations (suggesting consistency in metacognition across the upper part of the face recognition spectrum), they were largely driven by SR participants for target-present face-matching performance. This finding suggests that civilian top-performers may have greater insight into their face matching skills, and the SRQ may be particularly calibrated to discriminate between these individuals. However, even the largest effect sizes observed in this investigation were much milder than those from prosopagnosia studies, suggesting less utility for self-report in SRs screening program.

This conclusion is more strongly supported by the even milder effects observed in our second and third investigations, examining the use of self-report in police officers and civilians who had not been pre-selected according to their self-perceived face recognition skills. For the police officer sample, the SRQ was again a better predictor of face memory performance than single-item self-ratings. However, despite mild effect sizes in correlational analyses, the questionnaire showed little sensitivity in discriminating between SR and typical officers, and neither the SRQ nor single-item self-rating predicted target-present matching performance. Given the relatively stronger relationships in the civilian sample were largely driven by top-end performers (but for moderate correlations between self-report and matching performance in typical perceivers see *Shah et al., 2015a*), it is possible that the absence of the effect in police participants can be explained by the relatively lower proportion of SRs. Indeed, while our civilian sample all believed they possess superior face recognition skills, police officers were encouraged to participate regardless of their self-perceived face recognition ability. This interpretation is supported

by our third study, where no significant correlations were observed between subjective and objective face recognition performance in typical civilian participants.

Interestingly, the single-item ratings that were provided by "others" (i.e., family or friends for the top-end civilians, and colleagues for the officers) were mildly associated with both CFMT+ and target-present matching performance in both samples. This opens a potential role for a nomination system for SR screening, which may overcome any reluctance involved in self-referral. However, this relationship still only elicited a mild effect size, and a peer-nomination system would not be efficient for the identification of SRs in new recruits, given an individual would need to be observed "on-the-job" before a nomination could be made. Further, many roles within the police force do not provide the opportunity for an officer to demonstrate their face recognition skills, and their potential may subsequently be overlooked.

Interestingly, a mild effect size was also noted for the relationship between time "on-the-job" and target-absent matching performance in the police sample, with no associations observed with any self-report measure in police or civilian participants. It therefore seems likely that people rate their face recognition skills largely according to their successful target-present encounters, even on behavioral trait questionnaires. Pertinently though, previous work has dissociated target-present from target-absent performance in both typical perceivers (*Megreya & Burton, 2006*, *2007*) and SRs (*Bate et al., 2018*; *Bobak, Hancock & Bate, 2016*), supporting the hypothesis that self-report may be a better predictor of target-present performance. Thus, the findings reported here support previous work, and suggest that target-present and target-absent face recognition performance should be independently assessed in SR screening program.

## CONCLUSION

In sum, the work reported here is consistent with previous reports of only mild relationships between self-report measures and objective face recognition performance in the typical population. While we present the first behavioral trait questionnaire that is solely calibrated to detect top-end performance, this tool was only moderately useful in distinguishing between top-end performers, and of less value in randomly-selected populations. Importantly, self-report measures do not tap target-absent matching performance, and may be particularly unsuitable for the shortlisting of SR candidates within occupational settings.

### Funding

Sarah Bate is supported by a British Academy Mid-Career Fellowship (MD170004). The funders had no role in study design, data collection and analysis, decision to publish, or preparation of the manuscript.

## Grant Disclosure

The following grant information was disclosed by the authors:
British Academy Mid-Career Fellowship: MD170004.

## Competing Interests

The authors declare that they have no competing interests.

## Author Contributions

- Sarah Bate conceived and designed the experiments, performed the experiments, analyzed the data, contributed reagents/materials/analysis tools, prepared figures and/or tables, authored or reviewed drafts of the paper, approved the final draft.
- Gavin Dudfield contributed reagents/materials/analysis tools, authored or reviewed drafts of the paper, approved the final draft.

## Human Ethics

The following information was supplied relating to ethical approvals (i.e., approving body and any reference numbers):

Bournemouth University's Ethics Committee granted approval to carry out this study (application ID 11487).

## Data Availability

All raw data are available in the Supplemental File.

## Supplemental Information

Supplemental information for this article can be found online at http://dx.doi.org/10.7717/peerj.6330#supplemental-information.

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
