# Peer review of "Subjective assessment for super recognition: an evaluation of self-report methods in civilian and police participants"

_PeerJ, doi:10.7717/peerj.6330_

## Round 0.1 · original submission · Major Revisions

Two reviewers have read your paper and have many positive comments and a few important considerations that you should please take into account in your resubmission.

·

Basic reporting

See General comments

Experimental design

See General comments

Validity of the findings

See General comments

Additional comments

This paper describes two correlational studies examining the relationship between self-report measures and face identification ability. The first tests a group of people that self-identified as having high levels of face identification ability, the second recruits a cohort of police officers. Overall, the authors report mild-to-moderate correlations between a new self report measure called the SRU (developed by the authors) and performance in the face tests (CFMT, PMT), which is consistent with previous work on this topic.

The difference between the SRQ and other measures is that the authors specifically specifically designed it to discriminate between high performers. The authors find a statistically significant correlation between SRQ and face identification the first study (where the group mean performance is higher than an average group), but not it the second study using police officers. Because the police officers had normal levels of CFMT performance as a group, the authors conclude that the measure is effective in discriminating between high performers but not the general population.

Overall, this study addresses an important question which has clear practical and theoretical motivation. The use of a professional cohort in this study is a novel aspect of the design, and a real strength of the paper. Nevertheless, the evidence presented is somewhat ambiguous with respect to whether the SRQ is indeed better at discriminating between high performers. This question is central to the paper, and I suspect further data collection may be necessary to provide clarity on this question.

MAJOR POINT

The authors conclude that the SRQ is effective in discriminating between high performers but not the general population. However, based on the current data it is not clear whether the difference in SRU-CFMT correlation in Exp 1 and Exp 2 is caused by the fact that there are differences in the accuracy of these different groups or whether it has something to do with the professional experience/ context of the police officers. As the authors discuss (e.g. p5, para 2), there are many possible explanations for why this difference may emerge -- e.g., perhaps better performers have more insight into their abilities? Perhaps q'naires for professional cohorts need to ask about the tasks performed in their job rather than everyday life?. So the data as they stand are not especially informative with respect to this important question.

This could be addressed, either by stating more explicitly in the conclusion the need for future work to clarify this important question, or by doing that work and including in this paper.

Running a web-based study of individuals that are recruited through M-Turk or similar would enable the authors to examine the correlation between SRQ, PMT and CFMT+ in the general population. That would provide normative scores of the test in normal population, a group that have presented to the experimenters pertaining to have high levels of ability, and a professional cohort, which would strengthen the paper considerably regardless of the outcome.

MINOR

- Please provide summary descriptive statistics for the CFMT and PMT in Exp 1 and 2

- p4, Stolhoff et al citation: insert year

- p8/9: 'technical errors were monitored by the website' I wasn't sure what that meant

- p10: 'the sample all self-referred for super-recognition'. Please clarify what this means - e.g. did they visit the website following media about SRs? or did they e-mail in response to a recruitment ad? Please specify.

- p13 last sentence 'Unsurprisingly' ... why is this unsurprising?

- p14 section 'Subjective assessment for super recognition': clearer labelling of this section would help the reader. For example '19.3% of the variance' -- variance in what? Overall accuracy scores? In general I found this section quite confusing, and it would have helped if the predictor variables and the variables to-be-predicted were more explicitly stated.

- Table 5. Were these corrected for multiple comparison? It struck me that the authors devote a large section of the discussion to a correlation of .21 here (the relationship between time in police and target absent accuracy), and there is a very high risk of a false positive with an r value that low. I agree that this is a potentially interesting result but think the discussion should be prefaced by acknowledgment of this risk.

================
David White
UNSW Sydney

Reviewer 2 ·

Basic reporting

The article is well-written and easy to follow. The introduction in particular was a pleasure to read. The article includes sufficient background information and references to relevant literature. The article is structured in a logical way and figures are useful.

I will however note that the text refers to effect sizes in the tables that appear to be missing.

Experimental design

The article represents original, primary research. The research question is well defined, but in my opinion illogical (see General comments). The experimental design is robust, and the authors should be commended on their large police officer sample. Methods are reported in sufficient detail.

Validity of the findings

The conclusions drawn from the data and implications are not well-founded (see General comments).

Additional comments

MAJOR POINTS
The logic of the research question doesn’t make sense. The authors note that screening potential SRs with standardised tests can be very time consuming, and so it would be useful to have a quick way of detecting potential SRs. The authors then discuss how single-item self-report measures typically have very low predictive power and so while quick, don’t offer much useful information. I’m completely onboard up to this point. What I don’t understand is why, in response to this need for a quick and informative screening tool, the authors developed a 20-item self-report questionnaire. All the research points to people having poor insight into their face recognition abilities. Why not develop a screening tool that directly tests their face recognition ability? Why use a proxy for a skill you can actually test? I realise that the authors were trying to develop a quick screening tool, but in the time it takes to answer 20 questions on a 5-point scale participants could just as easily make multiple face identification decisions. If these comparisons were carefully selected based on performance data they would provide a reasonable basis upon which to screen for SRs. In fact, this approach is already used by at least 2 online SR tests. Given that the SRQ has almost no ability to screen SRs from ‘normal’ performers in civilian and police samples, I am at a loss as to why the authors are attempting to publish the SRQ at all. Surely the next logical (and necessary) step in this line of research is to develop a SR screening tool with the ability to predict SRs to a reasonable degree. I see the research reported in this paper as a first step in that line of work, and not necessarily worthy of publication in itself.

Could the authors please comment on the possibility that part of the reason why the SRQ had poor predictive power for the police officers was because they responded in a socially desirable way? It would be very easy for a participant to answer the SRQ as if they have very good face recognition ability even if they don’t. This is a critical issue with self-report measures and another reason to avoid using a proxy measure rather than measuring face recognition ability itself.

The general discussion largely ignores the fact the data show the SRQ has almost no discriminatory power for screening SRs, and makes claims as if the SRQ could be useful in applied settings. This sidesteps the data. It is not until the final sentence of the discussion (line 434-437) that the authors acknowledge that the SRQ is unlikely to be useful.

MINOR POINTS
Pg 3 line 70 – Phillips et al. paper is now published.
Pg 4 line 90 – Stollhoff citation missing year.
Pg 4 line 92 – This sentence is unclear. What values? What multiple validation studies?
Pg 5 line 114 – I don’t follow the authors’ argument that police officers could have more opportunities for feedback than civillians. In forensic situations the ground truth is often unknown and so feedback is rare. The type of feedback police officers receive often comes in the form of convictions, or peer-review, neither of which is veridical. As a percentage of the total number of identifications made, I would argue that civilians get more feedback.
Pg 8 line 174-177 – The description of the CFMT+ is very brief and not overly clear. The PMT was described in much more detail, even though a full description of both tests is available elsewhere.
Pg 10 line 224 (and elsewhere) – In a couple of places the authors write “…statistically significant difference in self-report between….”. Please help the reader by being clearer – self-report what?
Pg 10 line 238 – please clarify if 2SDs away from the SR mean in this context means ‘above’ or ‘below’.
Pg 10-11 – Is the SRQ only a significant predictor of the PMT when it’s broken down into target-present and absent trials?
Pg 11 line 257-259 – where are the effect sizes in table 3?
Pg 11 line 269 to Pg 12 line 274 – The summary overstates the predictive value of the SRQ.
Pg 14 line 342 to Pg 347 – The summary doesn’t provide important caveats on the claims, such as that some findings were only applicable to typicals, or SRs, or that the relationship with length of time as an officer was negative for SRs.
Figure 2 – panel A is missing the legend.

---

## Round 0.2 · accepted · Accept

The reviewer with previous major concerns is now satisfied that your publication is acceptable. Thank you for your diligence in updating your manuscript and congratulations!

# Reviewer 2 ·

Basic reporting

Please see general comments.

Experimental design

Please see general comments.

Validity of the findings

Please see general comments.

Additional comments

The authors have addressed all my comments satisfactorily and the additional experiment strengthens the paper considerably. I recommend publication.